# Glycolysis upregulation is neuroprotective as a compensatory mechanism in ALS

Ernesto Manzo[1], Ileana Lorenzini[2], Dianne Barrameda[1], Abigail G O'Conner[1], Jordan M Barrows[1], Alexander Starr[2], Tina Kovalik[2], Benjamin E Rabichow[2], Erik M Lehmkuhl[1], Dakotah D Shreiner[1], Archi Joardar[1], Jean-Charles Liévens[3], Robert Bowser[2], Rita Sattler[2], Daniela C Zarnescu[1,4,5]*

[1]Department of Molecular and Cellular Biology, University of Arizona, Tucson, United States; [2]Department of Neurology, Barrow Neurological Institute, Phoenix, United States; [3]Université de Montpellier, MMDN U1198, CC105, Montpellier, France; [4]Department of Neuroscience, University of Arizona, Tucson, United States; [5]Department of Neurobiology, University of Arizona, Tucson, United States

**Abstract** Amyotrophic Lateral Sclerosis (ALS), is a fatal neurodegenerative disorder, with TDP-43 inclusions as a major pathological hallmark. Using a *Drosophila* model of TDP-43 proteinopathy we found significant alterations in glucose metabolism including increased pyruvate, suggesting that modulating glycolysis may be neuroprotective. Indeed, a high sugar diet improves locomotor and lifespan defects caused by TDP-43 proteinopathy in motor neurons or glia, but not muscle, suggesting that metabolic dysregulation occurs in the nervous system. Overexpressing human glucose transporter GLUT-3 in motor neurons mitigates TDP-43 dependent defects in synaptic vesicle recycling and improves locomotion. Furthermore, *PFK* mRNA, a key indicator of glycolysis, is upregulated in flies and patient derived iPSC motor neurons with TDP-43 pathology. Surprisingly, *PFK* overexpression rescues TDP-43 induced locomotor deficits. These findings from multiple ALS models show that mechanistically, glycolysis is upregulated in degenerating motor neurons as a compensatory mechanism and suggest that increased glucose availability is protective.

DOI: https://doi.org/10.7554/eLife.45114.001

*For correspondence: zarnescu@email.arizona.edu

Competing interests: The authors declare that no competing interests exist.

## Introduction

Amyotrophic lateral sclerosis (ALS) is a neurodegenerative disease affecting upper and lower motor neurons resulting in progressive muscle weakness and eventual death (*Hardiman et al., 2017*). Cellular metabolic defects arise early, and contribute to the clinical manifestation of systemic defects that include weight loss, dyslipidemia, and hypermetabolism (reviewed in *Dupuis et al., 2011*; *Joardar et al., 2017*). However, it is unclear how metabolic defects relate, or may contribute to motor neuron degeneration.

Observations from both patients and animal models have begun to elucidate the mechanisms that contribute to metabolic defects in ALS. Interestingly, the human brain relies on glucose as its major energy source, and consumes ~20% of the body's glucose derived energy (*Mergenthaler et al., 2013*). Glucose metabolism defects have been previously reported in the frontal lobe and the cortex of ALS patients (*Ludolph et al., 1992*), and in mutant SOD1 mouse models of ALS (*Browne et al., 2006*; *Miyazaki et al., 2012*). Increasing glucose availability has been shown to reduce protein misfolding and delay neuronal degeneration in *C. elegans* models of neurodegeneration (*Tauffenberger et al., 2012*). In a recent pilot study, a high caloric diet based on high carbohydrate content has been shown to be well tolerated by patients, and reduce serious adverse events in ALS patients (*Wills et al., 2014*). Taken together, these findings suggest that although defective

glucose metabolism is still poorly understood in ALS, there lies a great opportunity to better understand its relationship to disease and explore its potential as a therapeutic avenue.

Our lab has developed a *Drosophila* model of ALS based on overexpression of human TDP-43 that recapitulates multiple disease aspects including cytoplasmic aggregates, neuromuscular junction (NMJ) abnormalities, lifespan, and locomotor defects (*Estes et al., 2011*; *Estes et al., 2013*). Importantly, the large majority of ALS patients (>97%) harbor TDP-43 cytoplasmic aggregates regardless of etiology, highlighting the importance of understanding TDP-43 toxicity mechanisms (*Ling et al., 2013*; *Neumann et al., 2006*). Here, we report that in a *Drosophila* model of TDP-43 proteinopathy, glycolytic metabolites and genes are altered, and are consistent with increased glucose consumption. Moreover, key genes responsible for driving glucose metabolism are upregulated in both patient derived induced pluripotent stem cell (iPSC) motor neurons and postmortem patient spinal cord tissue. In a fly model of TDP-43 proteinopathy, increased dietary glucose improves locomotor function and increases lifespan when TDP-43 is expressed in the central nervous system but not muscles. Additionally, genetic over-expression of human glucose transporters improves locomotor function, mitigates neuromuscular junction defects, and improves lifespan in a variant dependent manner. Finally, motor neuron overexpression of *PFK*, the rate limiting enzyme in glycolysis, improves locomotor function, suggesting that upregulation of glycolysis is neuroprotective through a compensatory mechanism in ALS. Together, our results show that increased glucose availability protects motor neurons and improves overall outcome in models of TDP-43 proteinopathy.

## Results

### Glycolysis and pentose phosphate pathways are altered by TDP-43 expression in motor neurons

To decipher metabolic alterations in ALS, we employed a *Drosophila* model of TDP-43 proteinopathy that recapitulates key features of the human disease including locomotor dysfunction and reduced survival (*Estes et al., 2011*; *Estes et al., 2013*). We conducted metabolomic profiling in whole third instar larvae expressing TDP-43 in motor neurons (D42 >TDP-43$^{WT}$ or D42 >TDP-43$^{G298S}$) and controls (D42>w$^{1118}$). These experiments showed significantly increased phosphoenolpyruvate (PEP) and pyruvate in both TDP-43$^{WT}$ and TDP-43$^{G298S}$ suggesting an increase in glucose metabolism (PEP, P$_{value}$ = 0.034 and pyruvate, P$_{value}$ = 0.007 for TDP-43$^{WT}$; PEP, P$_{value}$ = 0.002 and pyruvate, P$_{value}$ = 0.009 for TDP-43$^{G298S}$; see *Figure 1* and *Supplementary file 1*). Interestingly, pyruvate does not appear increased when endogenous TDP-43 is knocked down by RNAi in motor neurons suggesting that this alteration is caused by TDP-43 proteinopathy (*Figure 1—figure supplement 1*). In larvae expressing TDP-43$^{G298S}$, we also found a decrease in ribulose/xylulose 5-phosphate (P$_{value}$ = 0.011) in conjunction with increased sedoheptulose 7-phosphate (P$_{value}$ = 0.008) suggesting that increased glycolytic input into the Pentose Phosphate Pathway occurs in a variant dependent manner (see *Figure 1*, *Supplementary file 1*). Taken together these findings suggest that glucose metabolism is altered in the context of TDP-43 proteinopathy, consistent with previous reports of metabolite alterations such as increased pyruvate in plasma isolated from ALS patients (*Lawton et al., 2012*).

### Key glycolysis and pentose phosphate pathway enzymes are upregulated by TDP-43 expression in motor neurons

An increase in glycolytic metabolites suggests a possible increase in glycolysis. To test this hypothesis, we profiled the expression of multiple enzymes known to regulate flux through glycolysis and the Pentose Phosphate Pathway. Phosphofructokinase-1 (*PFK1; PFK* in *Drosophila*) levels are considered to be rate-limiting and control the rate of glycolysis, thus *PFK* expression provides a reliable determinant of flux for glucose breakdown (*Tanner et al., 2018*).

To measure the levels of *PFK* in the context of TDP-43 proteinopathy we performed qPCR in ventral nerve chords dissected from third instar larvae expressing TDP-43$^{WT}$ or TDP-43$^{G298S}$ in motor neurons (D42 >TDP-43) and w$^{1118}$ controls. These experiments showed that *PFK* transcript levels are increased in the context of TDP-43$^{WT}$ or TDP-43$^{G298S}$, albeit statistical significance was only achieved in the mutant TDP-43 (37% increase, P$_{value}$ = 0.06 for TDP-43$^{WT}$ and 59% increase, P$_{value}$ = 0.014 for

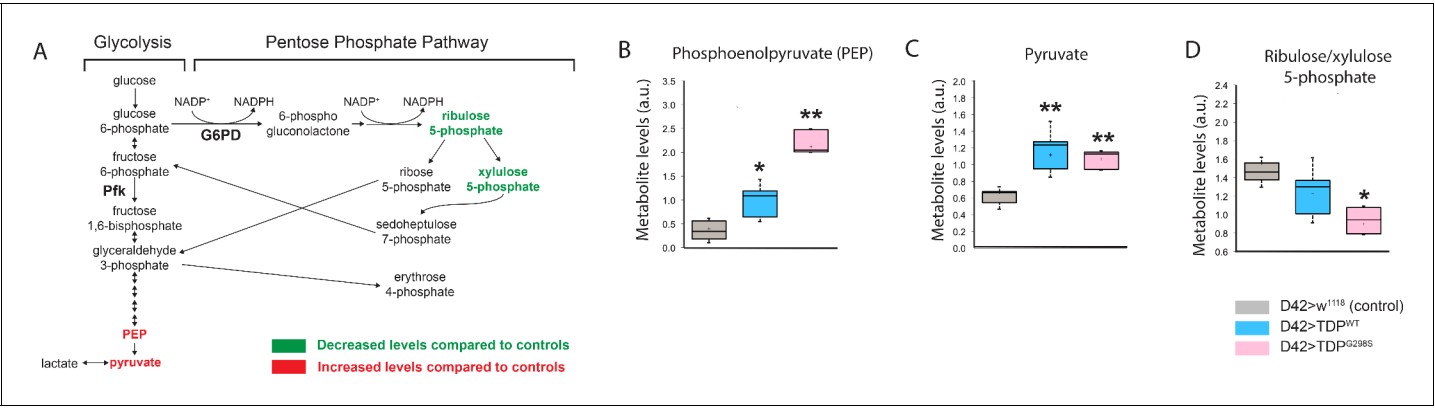

**Figure 1.** Glycolysis and pentose phosphate pathways are altered by TDP-43 expression in motor neurons. (**A**) Metabolite changes in glycolysis for whole larvae expressing TDP-43[WT] or TDP-43[G298S] were analyzed using mass spectrometry (see Materials and methods). Green and red font represent metabolites that are significantly changed compared to controls (w[1118]), as indicated. PEP and pyruvate were upregulated in both TDP-43[WT] and TDP-43[G298S] expressing flies. Changes in the pentose phosphate pathway metabolites are specific to larvae expressing TDP-43[G298S]. (**B, C, D**) Significant changes in select metabolites shown as box and whisker plots. Whiskers represent maximum and minimum values. Box edges represent upper and lower quartiles. Median values are denoted by horizontal lines within each box. One-way ANOVA was used to identify metabolites that differed significantly between experimental groups (N = 5).
DOI: https://doi.org/10.7554/eLife.45114.002

The following source data and figure supplements are available for figure 1:

**Source data 1.** Metabolic analysis.
DOI: https://doi.org/10.7554/eLife.45114.005

**Figure supplement 1.** Pyruvate measurements in whole larvae expressing RNAi knock-down constructs for the endogenous *Drosophila* TDP-43 (TBPH) with the D42 GAL driver show no significant changes compared to w[1118] controls.
DOI: https://doi.org/10.7554/eLife.45114.003

**Figure supplement 1—source data 1.** Pyruvate assay.
DOI: https://doi.org/10.7554/eLife.45114.004

TDP-43[G298S]; see *Figure 2A*). These results suggest that flux through glycolysis is significantly increased in motor neurons in the context of TDP-43[G298S].

Finally, glucose-6-phosphate dehydrogenase (*G6PD; Zw* in *Drosophila*) was selected due to its role as the commitment step for the pentose phosphate pathway (PPP). *Zw* transcript levels are increased specifically in TDP-43[G298S] (37% increase, $P_{value}$ = 0.012, see *Figure 2B*) but not TDP[WT], consistent with increased glycolytic influx through the PPP in the context of mutant TDP-43.

## *PFK* transcript is upregulated in human tissue

To confirm that the changes in our *Drosophila* model of ALS have clinical relevance, we assayed the transcript levels of *PFK* in human tissues and patient derived iPSC motor neurons. Humans have three isoforms of *PFK*, namely *PFKM*, *PFKL*, and *PFKP*. Although all are present in the brain, *PFKL* is expressed at lower levels (Human Brain Transcriptome: http://hbatlas.org/) thus we focused on *PFKP* and *PFKM*. For human tissues we used spinal cords isolated from ALS patients with TDP-43 proteinopathy and controls lacking neurological defects (see *Supplementary file 2* for demographic information). qPCR experiments show that both *PFKP* and *PFKM* levels are significantly increased in human spinal cords (*PFKP*: 80% increase, $P_{value}$ <0.001; *PFKM*: 55% increase, $P_{value}$ = 0.014, *Figure 2C*). iPSC motor neurons were derived from a patient harboring the TDP-43[G298S] mutation and differentiated as previously described (*Coyne et al., 2017*). qPCR profiling from iPSC derived motor neurons differentiated for 69–77 days (three independent differentiations, see *Supplementary file 3*) showed a 58% increase in *PFKP* ($P_{value}$ = 0.010; see *Figure 2D*) and a 45% increase in *PFKM* ($P_{value}$ = 0.036). These experiments show that glycolysis is also upregulated in patient spinal cords and motor neurons consistent with our findings in the fly model. Similarly, *G6PD* transcript levels are increased by 49% in spinal cords of ALS patients ($P_{value}$ = 0.015; see *Figure 2C*) and by 52% in patient derived motor neurons ($P_{value}$ = 0.012: see *Figure 2D*).

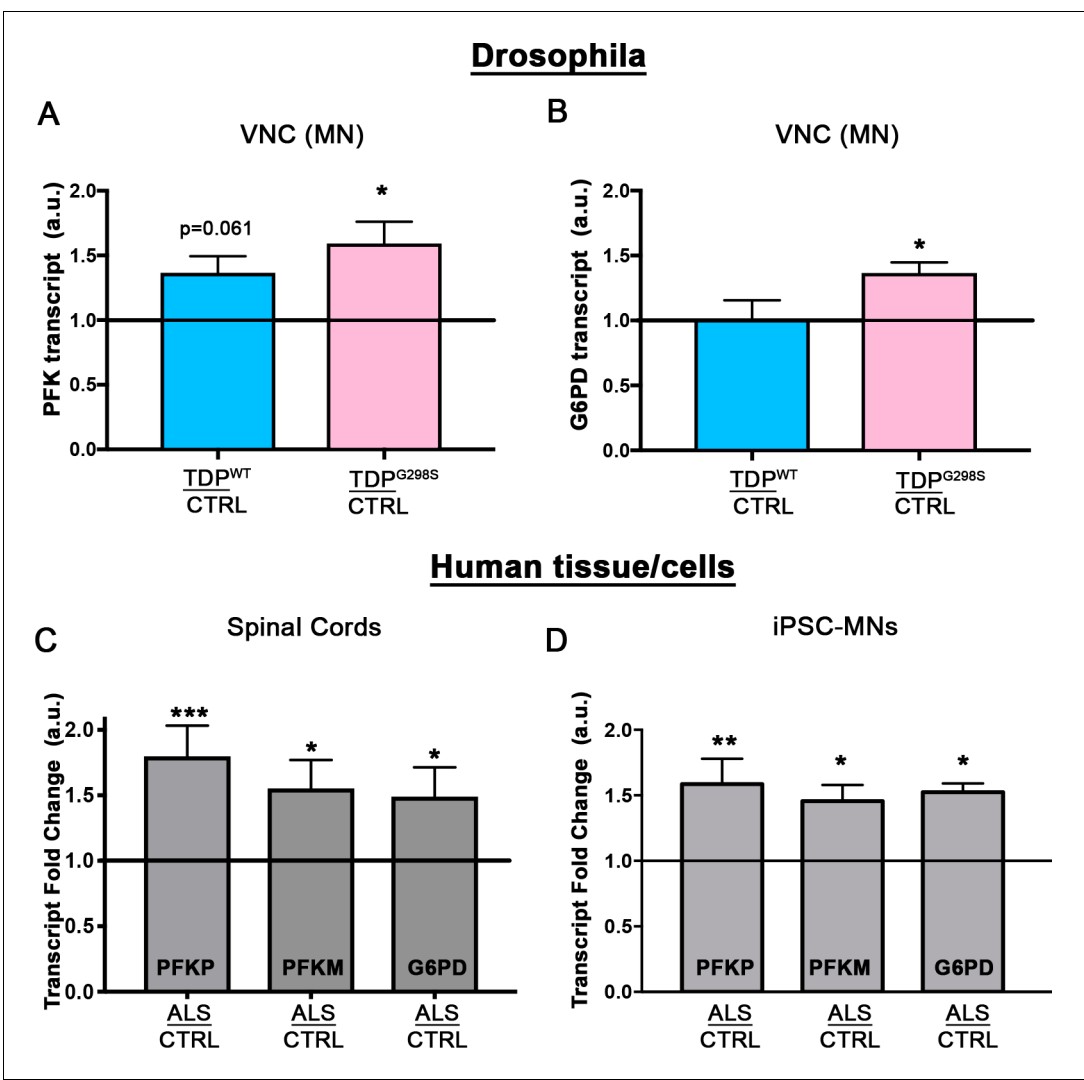

**Figure 2.** Glycolytic enzymes are transcriptionally upregulated. qPCR profiling of *PFK* (**A**; N = 5) and *G6PD* (**B**; N = 5) from ventral nerve cords of *Drosophila*. Human *PFKP*, *PFKM*, or *G6PD* mRNA levels were profiled in either spinal cords (**C**; N = 8 control and 9 ALS cases) or human iPSCs (**D**; N = 3 differentiations). Kruskal-Wallis test was used to identify significance.

DOI: https://doi.org/10.7554/eLife.45114.006

The following source data is available for figure 2:

**Source data 1.** *Drosophila* qPCR, ipsc neuron qPCR, and human spinal cord qPCR.

DOI: https://doi.org/10.7554/eLife.45114.007

## TDP-43 proteinopathy alters the capacity of motor neurons to import glucose

Our findings that TDP-43 expression in motor neurons alters glycolysis could be explained by an effect on glucose flux. To evaluate the levels of glucose in motor neurons, in the context of TDP-43 proteinopathy, we used a genetically encoded, FRET based, glucose sensor, FLII12Pglu-700μδ6 (*Fehr et al., 2003*; *Takanaga et al., 2008*). The sensor comprises a bacterial glucose-sensing domain and a pair of cyan and yellow fluorescent proteins that upon glucose binding, undergoes a conformational change and allows FRET to occur (*Figure 3A*). This genetically encoded sensor has been previously used in conjunction with the GAL4-UAS system in *Drosophila* to detect glucose flux and distribution in various cell types within the fly ventral nerve cord (*Volkenhoff et al., 2018*).

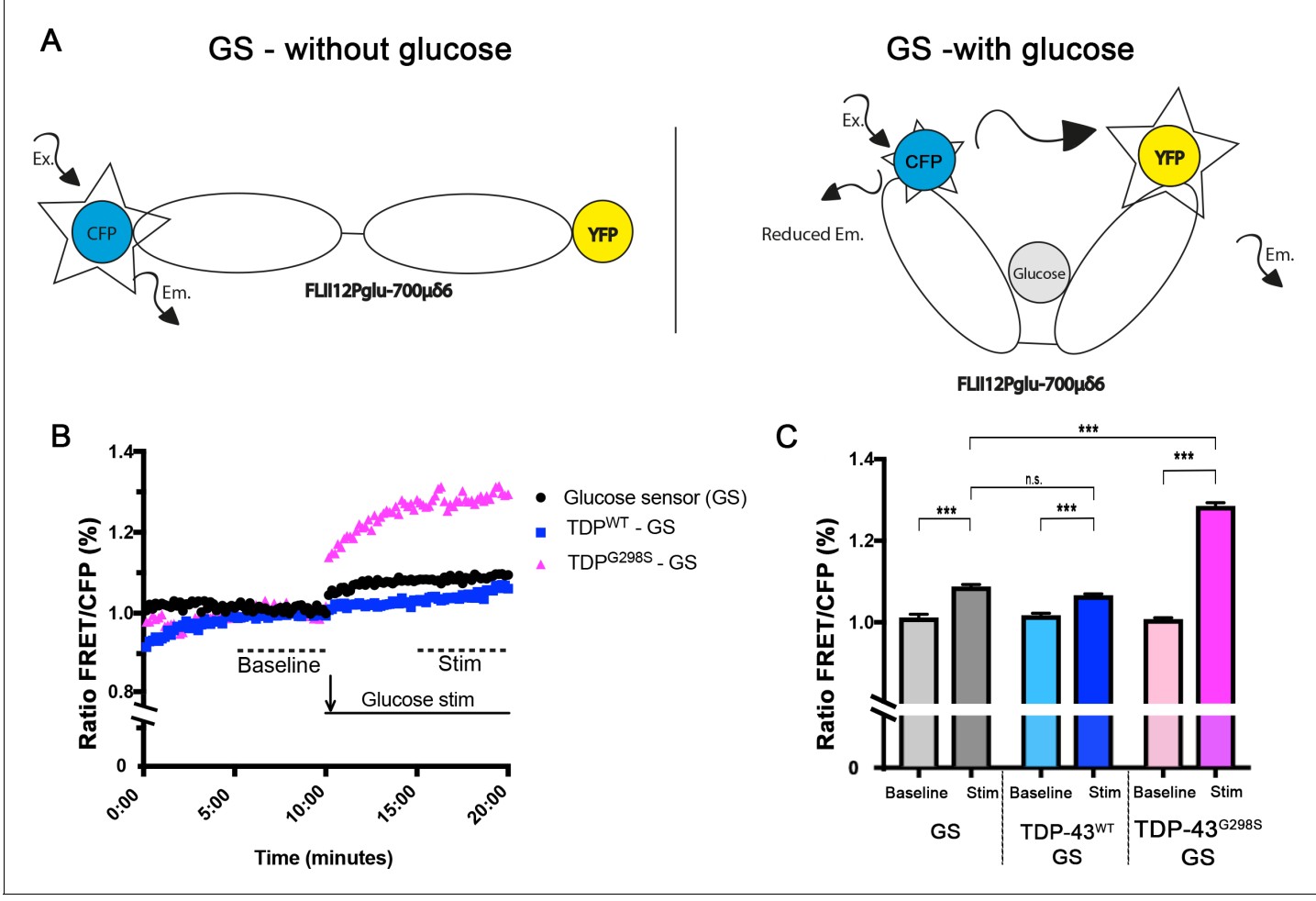

**Figure 3.** TDP expressing neurons have altered capacity to import glucose. FRET based glucose sensor described in *Volkenhoff et al. (2018)* was used to measure the glucose import capacity. Glucose sensor schematic described in (A). Ex – Excitation; Em – Emission. (B, C) TDP-43 expressing neurons and controls were imaged to detect CFP and FRET signal. 12–14 neurons were imaged every 10 s for 20 min. Values shown are the mean of 12–14 individual cells (ROI) from two ventral nerve cords (B). Mean values for 5–10 min and 15–20 min time intervals were used to calculate the 'baseline' and 'stimulated' ('Stim') values respectively (C). Kruskal-Wallis test was used to calculate significance.

DOI: https://doi.org/10.7554/eLife.45114.008

The following source data and figure supplement are available for figure 3:

**Source data 1.** Glucose sensor data.

DOI: https://doi.org/10.7554/eLife.45114.010

**Figure supplement 1.** Raw images of glucose sensor (A) or glucose sensor in the context of TDP-43[G298S] (B).

DOI: https://doi.org/10.7554/eLife.45114.009

Upon stimulation with glucose, neurons expressing the glucose sensor alone or together with TDP-43 showed a small but significant increase in the FRET/CFP ratio (*Figure 3B,C*). TDP-43[WT] expressing neurons showed no significant difference in the FRET/CFP signal compared to controls (*Figure 3B,C*, $P_{value}$ <0.99). In contrast, TDP-43[G298S] expressing neurons showed a significant increase in the FRET/CFP signal compared to controls (*Figure 3B,C*, $P_{value}$ <0.001). These results suggest that glucose flux is specifically enhanced in TDP-43[G298S] expressing neurons, and further substantiate our findings that glycolysis and the pentose phosphate pathway are significantly increased in the context of TDP-43 proteinopathy.

## A high glucose diet mitigates neuronal and glial induced TDP-43 toxicity

The alterations in glucose metabolism we have uncovered here could be a direct consequence of TDP-43 proteinopathy, or alternatively, could reflect a coping mechanism. To distinguish between these two scenarios, we fed TDP-43 expressing flies a high glucose diet. As we have previously shown, the neuronal or glial expression of TDP-43$^{WT}$ or TDP-43$^{G298S}$ caused locomotor defects as indicated by increased larval turning time compared to controls (*Estes et al., 2011*; *Estes et al., 2013*). Of note, a high glucose diet increased the larval turning time of control larvae, consistent with previous reports that high sugar intake induces insulin resistance and obesity in flies (*Musselman et al., 2011*; *Na et al., 2013*). In contrast, a high glucose diet improved the larval locomotor defect caused by either neuronal, or glial expression of TDP-43 (*Figure 4A* and *Figure 4—figure supplement 1*). Interestingly, we found that increasing the concentration of glucose five-fold or higher rescued locomotor phenotypes caused by TDP-43 while no improvement was observed when glucose is only increased by two-fold (*Figure 4—figure supplement 1*). Next, we examined the effect of glucose on lifespan and found that the reduced survival caused by neuronal or glial expression of TDP-43$^{WT}$ or TDP-43$^{G298S}$ was also significantly improved by feeding *Drosophila* a high sugar (HS) versus a regular sugar (RS) diet (23.5 days for TDP-43$^{WT}$ on RF and 53 days for TDP-43$^{WT}$ on HS, $P_{value}<0.0001$; 14.5 days for TDP-43$^{G298S}$ on RS and 41 days for TDP-43$^{G298S}$ HS, $P_{value}<0.0001$; *Figure 4B*, *Figure 4—figure supplement 2*). Together, these results suggest that while a high sugar diet is detrimental to control flies, it improves key hallmarks of motor neuron disease including locomotor dysfunction and reduced survival in ALS flies.

## Overexpression of human glucose transporters mitigate TDP-43 dependent phenotypes in neurons or glia

Our results on a high sugar diet suggest that increased glucose availability is protective in our fly model of ALS. However, because feeding can cause systemic effects, and to further probe the role of glucose in our *Drosophila* model of ALS, we co-expressed TDP-43 and the human glucose transporter GLUT-3 in motor neurons or glia. GLUT-3 is the primary glucose transporter of mammalian neurons, and has a high capacity for glucose (*Ferreira et al., 2011*). Co-expression of GLUT-3 and TDP-43$^{WT}$ or TDP-43$^{G298S}$, in either motor neurons or glia cells, improved the locomotor deficits caused by TDP-43 expression alone (*Figure 4C*). Moreover, GLUT-3 expression in neurons improved the locomotor ability of adult flies expressing TDP-43$^{WT}$ or TDP-43$^{G298S}$ (as measured by negative geotaxis assays, *Figure 4—figure supplement 3*). When using the standard log-rank test for analyzing the survival data, lifespan was significantly increased when GLUT-3 was co-expressed with TDP-43$^{G298S}$ in motor neurons, but not glial cells (18 days for TDP-43$^{G298S}$ and 30 days for TDP-43$^{G298S}$, $P_{value} =<0.0001$). However, when using the Mantel-Cox test, which weighs earlier deaths more significantly, we found that GLUT-3 overexpression increases lifespan when TDP-43$^{G298S}$ is expressed in glial cells (43.5 days vs 51.5 days, $P_{value} = 0.015$). GLUT-3 had no effect on lifespan when TDP-43$^{WT}$ was expressed in either motor neurons or glia. Importantly, GLUT-3 overexpression had no effect on TDP-43 suggesting that its protective effect occurs through a mechanism independent of TDP-43 levels (see *Figure 4—figure supplement 4*). Together, these findings suggest that increased glucose availability is protective in both motor neurons and glia but to a different extent that likely reflects different metabolic capabilities of the two cell types.

Although GLUT-3 is the predominant glucose transporter in the brain, recently, the human glucose transporter GLUT-4, an insulin dependent glucose transporter expressed primarily in muscle has been shown to localize to the plasma membrane to support synaptic activity (*Ashrafi et al., 2017*). To test whether GLUT-4 also modulates TDP-43 proteinopathy, we co-expressed GLUT-4 and TDP-43 in either motor neurons or glia. Similar to GLUT-3, co-overexpression of GLUT-4 and TDP-43$^{WT}$ or TDP-43$^{G298S}$, in either motor neurons or glial cells, improved the locomotor deficits caused by TDP-43 expression alone (*Figure 4—figure supplement 5*). Notably, while TDP-43 overexpression in muscle causes larval turning phenotypes, co-overexpression of either GLUT-3 or GLUT-4 in muscle does not mitigate TDP-43 dependent locomotor deficits, suggesting that the primary metabolic defect lies in the nervous system (*Figure 4—figure supplements 2* and *5*).

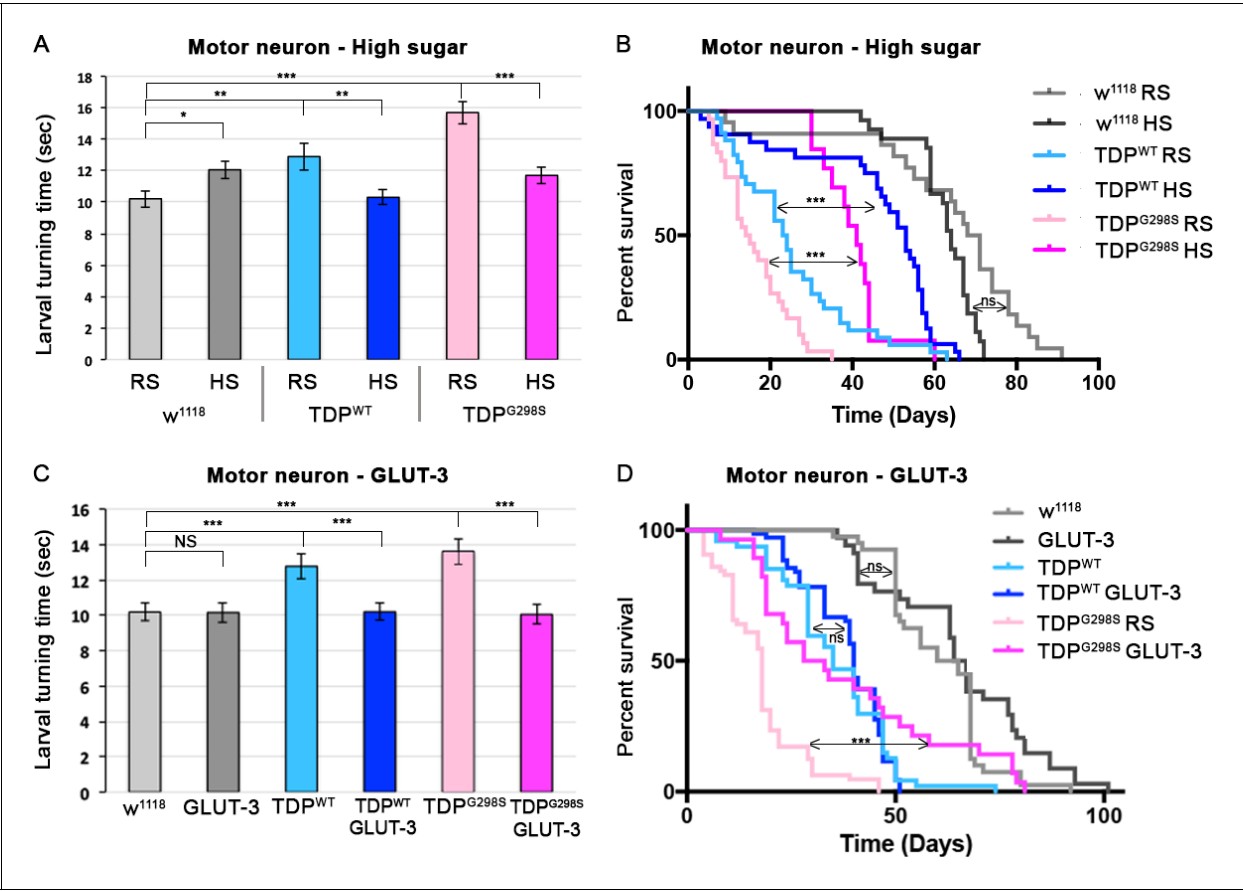

**Figure 4.** A high glucose diet rescues neuronal TDP-43 toxicity in flies. TDP-43[WT] or ALS associated TDP-43[G298S] were expressed in MNs (using GAL4-UAS). (**A, B**) Larval turning and lifespan assays for *Drosophila* fed a cornmeal based food containing either regular concentration of sugar (RS) or a high sugar diet (HS:10x the standard amount of sugar). (**C, D**) Larval turning and lifespan assays for *Drosophila* expressing GLUT-3 on its own or with TDP-43, as indicated. At least 30 larvae were tested in larval turning assays and on average 20 adults were assayed for survival. Kruskal-Wallis test and Log-rank (Mantel-Cox) test was used to determine statistical significance for larval turning and survival curve respectively. * - p<0.05, ** - p<0.01, *** - p<0.001.
DOI: https://doi.org/10.7554/eLife.45114.011

The following source data and figure supplements are available for figure 4:

**Source data 1.** D4210x, 5x, 2x, 1x, 0x Glucose Larval Turning D42 AO; GLUT 3LT; GLUT3 survival; and HS survival.
DOI: https://doi.org/10.7554/eLife.45114.021

**Figure supplement 1.** Larval turning assays on regular cornmeal-molasses food supplemented with various amounts of glucose (2X, 5X or 10X, as shown).
DOI: https://doi.org/10.7554/eLife.45114.012

**Figure supplement 1—source data 1.** Glucose Larval Turning.
DOI: https://doi.org/10.7554/eLife.45114.013

**Figure supplement 2.** A high sugar diet or GLUT-3 overexpression are partially protective when TDP-43 is expressed in glia but not in muscles.
DOI: https://doi.org/10.7554/eLife.45114.014

**Figure supplement 2—source data 1.** (i) Glia HS LT; (ii) Glia HS SC; (iii) Glia Glut3 LT; (iv) GLUT3 Glia SC; (v) Muscle HS LT; (vi) Muscle Glut3 LT.
DOI: https://doi.org/10.7554/eLife.45114.015

**Figure supplement 3.** Negative geotaxis assay on adult flies expressing TDP-43[WT] or TDP-43[G298S] alone or in conjunction with GLUT-3.
DOI: https://doi.org/10.7554/eLife.45114.016

**Figure supplement 4.** GLUT-4 overexpression mitigates locomotor defects when TDP-43 is expressed in motor neurons or glia but not muscles.
DOI: https://doi.org/10.7554/eLife.45114.017

**Figure supplement 4—source data 1.** Western Blot Quantifications.
DOI: https://doi.org/10.7554/eLife.45114.018

**Figure supplement 5.** Larval turning assays for GLUT-4 and TDP-43 overexpression in motor neurons (i), glia (ii) or muscles (iii).
DOI: https://doi.org/10.7554/eLife.45114.019

**Figure supplement 5—source data 1.** GLUT 4 Glia LT; GLUT4 MN LT; and GLUT4 Muscle LT.

*Figure 4 continued on next page*

*Figure 4 continued*

DOI: https://doi.org/10.7554/eLife.45114.020

## TDP-43 dependent defects at the neuromuscular junction are rescued by GLUT-3

We previously described defects at the *Drosophila* neuromuscular junction (NMJ) caused by neuronal expression of TDP-43. These defects include a reduction in synaptic bouton number (*Coyne et al., 2014*; *Estes et al., 2013*) and a reduction in synaptic vesicle endocytosis as indicated by FM1-43 dye uptake experiments (*Coyne et al., 2017*). To test whether increased GLUT-3 could also rescue morphological and functional defects at the larval NMJ, we co-expressed GLUT-3 and TDP-43 in motor neurons. Co-expression of GLUT-3 rescued NMJ size as indicated by the total number of synaptic boutons in both TDP-43$^{WT}$ ($P_{value}$ = 0.004) and TDP-43$^{G298S}$ ($P_{value}$ = 0.023) expressing larvae (*Figure 5A and C*). Moreover, GLUT-3 overexpression partially restores synaptic vesicle endocytosis deficits caused by TDP-43 (*Figure 5B and D*, $P_{value}$ <0.001 for both TDP-43$^{WT}$ and TDP-43$^{G298S}$). Taken together, these results show that GLUT-3 expression in motor neurons mitigates morphological and functional defects caused by TDP-43 at the NMJ and suggest that increased glucose availability in motor neurons is neuroprotective in ALS.

## Neuronal co-expression of *PFK* rescues TDP-43 induced toxicity

Increased *PFK* levels in conjunction with our findings that increased glucose availability is protective suggest that increased glycolysis may be compensatory. To test this hypothesis, we directly manipulated glycolysis genetically, either via *PFK* overexpression or RNAi knock-down, in the context of TDP-43 proteinopathy. Overexpression of *PFK* alone in motor neurons negatively altered locomotor activity suggesting that in normal neurons high glycolytic activity may be toxic (*Figure 6A*). However, the co-expression of *PFK* with TDP-43$^{WT}$ or TDP-43$^{G298S}$ rescued locomotor defects to control levels ($P_{value}$ = 0.003 for TDP-43$^{WT}$ and $P_{value}$ <0.001 for TDP-43$^{G298S}$, *Figure 6A*). In contrast, *PFK* RNAi knock-down had no significant effect on larval turning when expressed in motor neurons alone (53% knock-down, $P_{value}$ = 0.04, see Materials and methods). However, when *PFK* was knocked down by RNAi in the context of TDP-43 proteinopathy we found that it significantly increased larval turning times for TDP-43$^{WT}$ albeit it had no effect on TDP-43$^{G298S}$ ($P_{value}$ <0.001 for TDP-43$^{WT}$ and $P_{value}$ = 0.767 for TDP-43$^{G298S}$, *Figure 6B*). One possible interpretation for this variant dependent effect is that the metabolic deficit of TDP-43$^{G298S}$ is so severe that it may be difficult to exacerbate by further altering energy metabolism. Of note, a second *PFK* RNAi line (y(1) sc[*] v(1) P{y[+t7.7] v [+t1.8]=TRiP.HMS01324}attP2) was tested, but was lethal when expressed in motor neurons.

Similar to the experiments with motor neurons, the over-expression of *PFK* in glial cells causes a locomotor defect, while the expression of two independent *PFK* RNAi lines has no effect on locomotor activity as determined by larval turning assays (*Figure 6—figure supplement 1*). When *PFK* is co-expressed in glial cells with TDP-43$^{WT}$ there is a slight but not statistically significant reduction in larval turning ($P_{value}$ >0.302) while it significantly mitigates TDP-43$^{G298S}$ ($P_{value}$ = 0.001, *Figure 6—figure supplement 1*). Moreover, the knockdown of *PFK* in conjunction with TDP-43$^{WT}$ significantly impaired locomotor function when compared to TDP-43$^{WT}$ expression alone ($P_{value}$ <0.001, *Figure 6—figure supplement 1*). *PFK* knock down did not affect TDP-43$^{G298S}$ expressing flies $P_{value}$ >0.057, *Figure 6—figure supplement 1*). Importantly, the protective effect of *PFK* overexpression in motor neurons does not occur via reducing TDP-43 levels (*Figure 6—figure supplement 1*). These results indicate that an increase in glycolysis, in either motor neurons or glial cells, can ameliorate TDP-43 toxicity.

## Compensatory upregulation of *PFK* transcript is not a consequence of sequestration in TDP-43 insoluble complexes

Our findings of increased *PFK* mRNA levels and genetic interaction data with *PFK* support a protective compensatory mechanism (see *Figure 7* for model). This could be in part explained by the ribostasis hypothesis (*Ramaswami et al., 2013*) whereby *PFK* mRNA associates with TDP-43 in insoluble complexes and as a result, transcription is upregulated to compensate for the amount of mRNA sequestered and unavailable for translation. This however, does not seem to be the case, as soluble

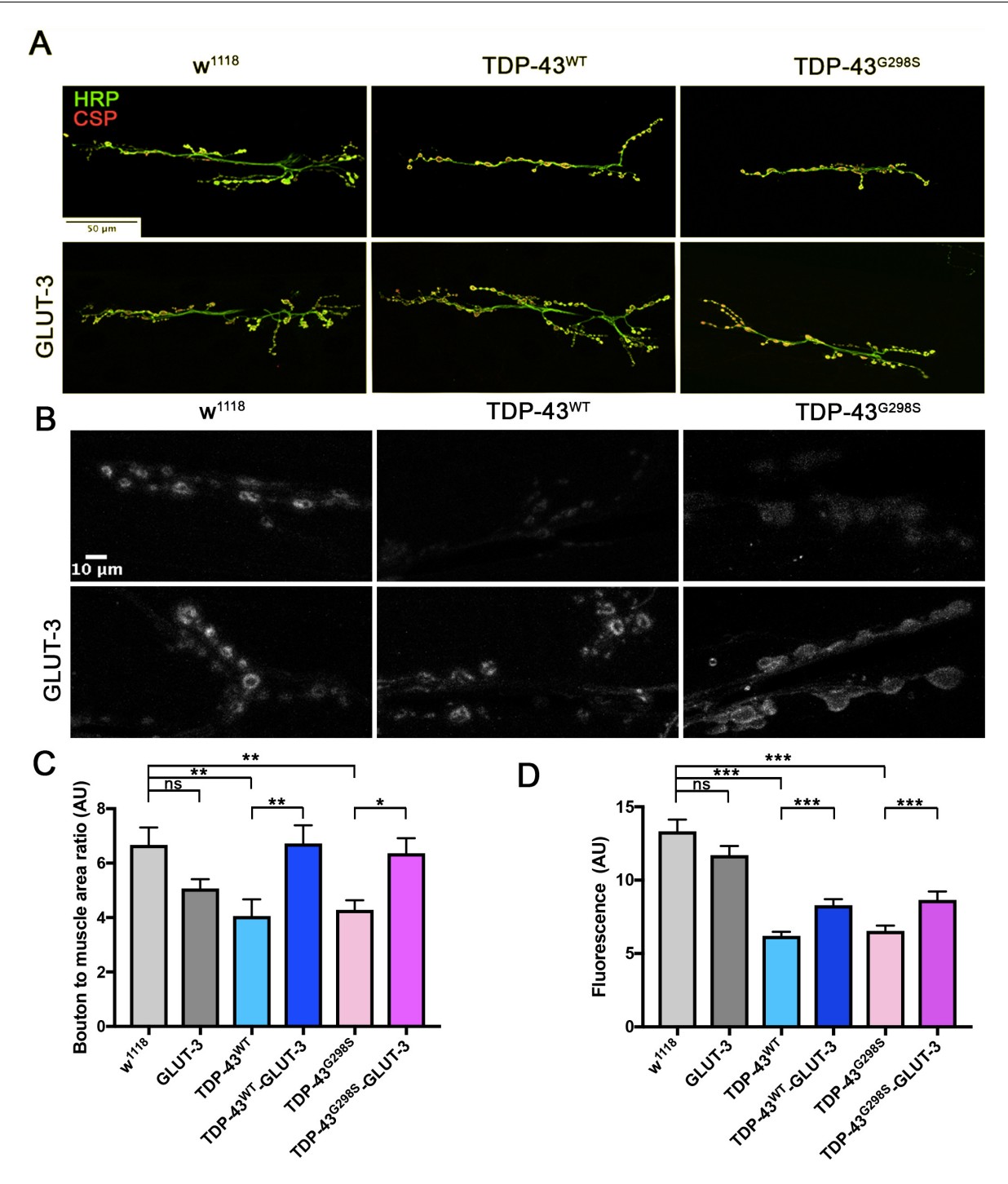

**Figure 5.** TDP-43 dependent defects at the NMJ are rescued by GLUT-3. Third instar larvae NMJ from segment A3, muscle 6/7 were immunostained for CSP and HRP (**A**) or analyzed for their ability to endocytose FM1-43 dye upon stimulation with 90 mM KCl (**B**). (**A, C**) Neuronal TDP-43 expression in *Drosophila* neurons reduces the number of boutons (labeled with CSP and HRP (**A, C**) and reduces FM1-43 dye uptake (**B, D**). These morphological (**A, C**) and functional (**B, D**) deficits are rescued by co-expression of GLUT-3. N = 7–10 larvae. Kruskal-Wallis test was used to identify significance.

DOI: https://doi.org/10.7554/eLife.45114.022

The following source data is available for figure 5:

**Source data 1.** Bouton count analysis and FM1-43 GLUT-3 analysis.
DOI: https://doi.org/10.7554/eLife.45114.023

versus insoluble fractionation experiments do not support this hypothesis (*Figure 7—figure supplement 1*). Surprisingly, *PFK* mRNA appears to be more soluble in the context of TDP-43$^{WT}$ versus TDP-43$^{G298S}$, consistent with different TDP-43 variants utilizing distinct pathomechanisms, as we have previously reported (*Coyne et al., 2017*; *Estes et al., 2013*).

## Discussion

It is widely known that ALS patients display gross metabolic dysregulation described as hypermetabolism (for a recent review, see *Joardar et al., 2017*). However, a major challenge in the field has been to understand how these clinical observations relate to metabolic changes at the cellular level. Here, we take advantage of cell type specific genetic tools available in *Drosophila* to pinpoint how specific metabolic changes in neurons and glia relate to disease progression. Using metabolic profiling we first found that the neuronal expression of TDP-43 increases the abundance of pyruvate, the end product of glycolysis. Notably, this alteration is consistent with metabolite changes in plasma from ALS patients (*Lawton et al., 2012*) and suggest an increase in glycolysis. Substantiating this scenario is transcriptional profiling of ALS spinal cords and patient derived iPSC motor neurons showing that just as in flies, *PFK*, the rate limiting step in glycolysis, is significantly upregulated (*Figure 2*). Interestingly, metabolic profiling in flies also suggests increased glycolytic input into the pentose phosphate pathway (*Supplementary file 1*), which provides a mechanism for countering oxidative stress via increased NADPH production (*Kruger and von Schaewen, 2003*). Supporting this possibility are our findings that *G6PD*, the rate limiting enzyme of the pentose phosphate pathway is upregulated in fly and human ALS ventral and spinal cords, respectively (*Figure 2*). Oxidative stress is a well-established ALS manifestation (*Barber and Shaw, 2010*) and the most recent FDA

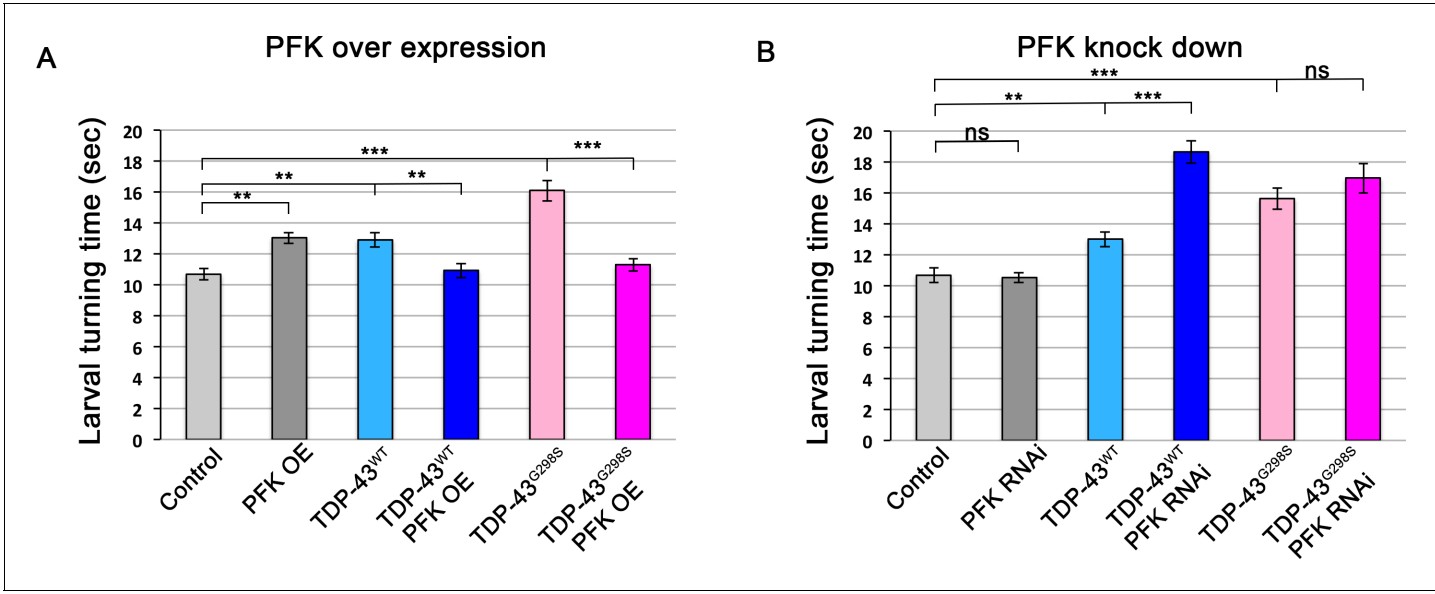

**Figure 6.** Co-overexpression of *PFK* rescues TDP-43 induced locomotor defects. (**A**) TDP-43$^{WT}$ or ALS associated TDP-43$^{G298S}$ were expressed in MNs (using the GAL4-UAS system together with *Drosophila* UAS-PFK). (**B**) TDP-43$^{WT}$ or ALS associated TDP-43$^{G298S}$ were expressed in MNs (using the GAL4-UAS system together with *Drosophila* UAS-PFK$^{RNAi}$). N = 30 larvae. Kruskal-Wallis was used to determine statistical significance. * - P $_{value}$ < 0.05, ** - P $_{value}$ < 0.01, *** - P $_{value}$ < 0.001.

DOI: https://doi.org/10.7554/eLife.45114.024

The following source data and figure supplements are available for figure 6:

**Source data 1.** *PFK* D42 LT.
DOI: https://doi.org/10.7554/eLife.45114.027

**Figure supplement 1.** Larval turning assays for *PFK* overexpression (**i**) or RNAi (**ii**) in the context of TDP-43 in glia.
DOI: https://doi.org/10.7554/eLife.45114.025

**Figure supplement 1—source data 1.** *PFK* Glia.
DOI: https://doi.org/10.7554/eLife.45114.026

approved drug for treating ALS patients, Radicava, is aimed at countering this aspect of the disease (*Cruz, 2018*). Collectively, molecular and metabolic alterations identified in the fly validate in patient derived motor neurons and spinal cords, albeit the magnitude of the changes found in human ALS tissues is often more dramatic and, at some level, more comparable with mutant TDP-43 dependent phenotypes in flies. These findings highlight the predictive power of the fly model and its relevance to studying disease pathomechanisms in humans.

Using a genetically encoded glucose sensor we show that TDP-43$^{G298S}$ expressing motor neurons have increased capacity to import glucose (*Figure 3*). This is consistent with increased glycolysis and its end product, pyruvate. It is unclear why despite finding that pyruvate is also increased in motor neurons overexpressing TDP-43$^{WT}$, we could not detect higher glucose uptake than in controls, although this is consistent with *PFK* transcript levels trending up, yet not reaching significance in our data set (*Figure 2*). A possible explanation is that while our model is based on TDP-43 expression specifically in motor neurons or glia, the metabolite profiling was performed on whole larvae and captured non-cell autonomous changes, occurring in cells other than motor neurons. Nevertheless, taken together our data show that there is a clear dysregulation of glycolysis in TDP-43 expressing *Drosophila* and patient tissue samples. Alterations in glucose metabolism may also be a feature of multiple ALS types. Indeed, pre-symptomatic SOD1 mutant mice have been shown to have high glucose levels in spinal cords; however, glucose levels gradually decrease as disease symptoms progress (*Miyazaki et al., 2012*). One question that remained is whether alterations in glucose metabolism are compensatory or causative.

In an effort to address this question, we first increased glucose availability in *Drosophila* by raising the dietary concentration of glucose (5–10 X higher than normal levels in standard fly food), which mitigates locomotor defects and increases lifespan in ALS flies while making wild-type larvae sluggish (*Figure 3*). Interestingly, a high sugar diet only mitigates locomotor defects when TDP-43 is expressed in motor neurons or glia but not muscles, indicating that the primary metabolic defect lies in the nervous system and that muscles do not benefit from increased glucose availability (*Figure 4* and *Figure 4—figure supplements 2* and *5*). This is somewhat surprising given that muscles normally rely on glycolysis for ATP production but is also consistent with findings that in ALS, muscles switch to primarily using lipids and not glucose for energy production (*Palamiuc et al., 2015*).

This systemic approach, however, does not provide information on which cells specifically benefit from increased glucose availability. To answer this question, we intervened genetically by overexpressing human glucose transporters GLUT-3 or GLUT-4 in motor neurons, glia or muscles (*Figures 4* and *5* and *Figure 4—figure supplements 2* and *5*). These experiments showed that both GLUT-3

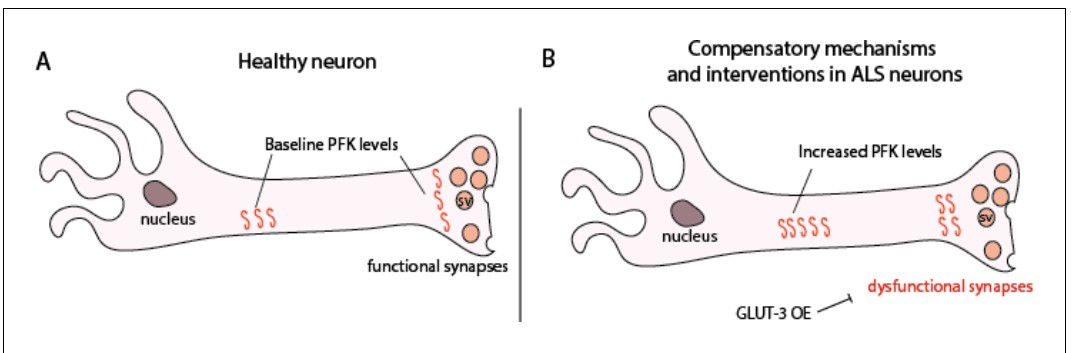

**Figure 7.** Proposed model showing *PFK* transcript levels increase in response to TDP-43 proteinopathy. (**A**) Neurons from non-diseased patients. (**B**) ALS neurons showing an increase in *PFK* transcript levels. SV – synaptic vesicle.

DOI: https://doi.org/10.7554/eLife.45114.028

The following source data and figure supplements are available for figure 7:

**Figure supplement 1.** Cellular fractionations from third instar larvae.
DOI: https://doi.org/10.7554/eLife.45114.029

**Figure supplement 1—source data 1.** Soluble fractionations.
DOI: https://doi.org/10.7554/eLife.45114.030

and GLUT-4 rescued locomotor dysfunction of TDP-43 expressing larvae in the nervous system, but not in muscles (*Figure 4* and *Figure 4—figure supplements 2* and *5*) confirming our findings from high sugar feedings. Furthermore, our findings that overexpression of either GLUT-3 or GLUT-4 can help mitigate specific ALS-like phenotypes in flies are consistent with recent reports that both GLUT-3 and GLUT-4 are required at synapses to regulate activity (*Ashrafi et al., 2017*; *Ferreira et al., 2011*). Together, our dietary intervention and GLUT-3/4 overexpression experiments indicate that increased glucose availability in motor neurons and glia confers protection in models of TDP-43 proteinopathy, in a cell type dependent manner.

A possible explanation for why increased glucose availability improves defects caused by TDP-43 proteinopathy is that cells are actively consuming glucose to increase glycolytic flux. A recent study using vertebrate cell lines suggests that glycolytic flux depends on four key steps. They include: (1) glucose uptake by glucose transporters, (2) hexokinase levels, (3) *PFK* levels (and associated enzymes), and (4) lactate export, but not other enzymes or steps involved in glycolysis (*Tanner et al., 2018*). The co-over-expression of *PFK* mitigates TDP-43 defects in flies, and is consistent with our GLUT-3 over-expression experiments (*Figure 6*). Surprisingly, *PFK* transcript levels were higher in human ALS post-mortem tissues compared to controls, however protein levels in total tissue homogenates were not significantly increased despite trending upwards (data not shown). Nevertheless, our findings of increased *PFK* mRNA levels in flies and human tissues, and the genetic interaction experiments support the notion that glycolysis is increased in ALS as a compensatory mechanism (see *Figure 7* for model).

The fact that neurons would compensate deficits in cellular energetics through increasing glycolysis makes sense in the light of well-established defects in mitochondrial function. TDP-43 has been previously shown to localize to the inner mitochondrial membrane of mitochondria (*Wang et al., 2016*), and it has been shown to cause abnormally small mitochondria in *Drosophila* (*Khalil et al., 2017*). Consistent with the importance of glycolysis in neurodegeneration, there are reports of SNPs within *PFK* that are present in ALS patients but not in healthy controls (*Rouillard et al., 2016*; *Xie et al., 2014*). Our findings that glycolysis is increased as a compensatory mechanism and this confers neuroprotection is consistent with findings from a small clinical trial showing that a high caloric high carbohydrate diet improves patient outcome (*Wills et al., 2014*) and also that patients with diabetes have a later onset of disease (*Kioumourtzoglou et al., 2015*). While upregulation of glycolysis decreases the dependence of the organism for ATP derived from mitochondria, it is also possible that increased glycolysis is improving organismal health through a different mechanism. For example, pyruvate has been previously shown to protect mitochondria from oxidative stress (*Wang et al., 2007*). Future studies should focus on the link between glycolysis, oxidative stress and mitochondria in ALS.

## Materials and methods

**Key resources table**

| Reagent type (species) or resource | Designation | Source or reference | Identifiers | Additional information |
|---|---|---|---|---|
| Genetic reagent (*D. melanogaster*) | w[1118];UAS;TDP[WT]-YFP | doi: 10.1242/dmm.010710 | TDP[WT] | Stock maintained by the Zarnescu Laboratory |
| Genetic reagent (*D. melanogaster*) | w[1118];UAS-TDP[G298S]-YFP | doi: 10.1242/dmm.010710 | TDP[G298S] | Stock maintained by the Zarnescu Laboratory |
| Genetic reagent (*D. melanogaster*) | w[1118];UAS-GLUT-3 | doi: 10.1371/journal.pone.0118765 | GLUT-3 | Stock maintained by the Liévens Laboratory |
| Genetic reagent (*D. melanogaster*) | UAS-HA-GLUT-4-GFP | doi: 10.1371/journal.pone.0077953 | GLUT-4 | Stock maintained by the Pick Laboratory |

*Continued on next page*

*Continued*

| Reagent type (species) or resource | Designation | Source or reference | Identifiers | Additional information |
|---|---|---|---|---|
| Genetic reagent (*D. melanogaster*) | UAS-FLII12Pglu-700µδ6 | doi: 10.1016/j.jinsphys. 2017.07.010 | Glucose Sensor | Stock maintained by the Schirmeier Laboratory |
| Genetic reagent (*D. melanogaster*) | Motor neuron - D42-GAL4 Driver | Bloomington | FBst0008816 | |
| Genetic reagent (*D. melanogaster*) | Repo- Glial GAL4 driver | Bloomington | FBti0018692 | |
| Genetic reagent (*D. melanogaster*) | Bg487 - Muscle driver | Bloomington | FBti0004407 | |
| Genetic reagent (*D. melanogaster*) | TBPH RNAi | Bloomington | FBst0039014 | |
| Genetic reagent (*D. melanogaster*) | TBPH RNAi | Bloomington | FBti0128633 | |
| Genetic reagent (*D. melanogaster*) | PFK over expression | Bloomington | FBst0060675 | |
| genetic reagent (*D. melanogaster*) | PFK RNAi | Bloomington | FBst0034336 | |
| genetic reagent (*D. melanogaster*) | PFK RNAi | Bloomington | FBst0036782 | Used as second RNAi line for glial expression |
| Antibody | DCSP-2 | DSHB | 6D6 | 1/300 |
| Antibody | Alexa Fluor 488 Phalloidin | ThermoFisher | A12379 | 1/200 |
| Antibody | Alexa Fluor 568 | Invitrogen | A10037 | 1/500 |
| Antibody | Anti-HRP -CY5 | Jackson Labs | 123-165-021 | 1/200 |
| Antibody | GFP Monoclonal | Takara | 6326375 | 1/20000 |
| Antibody | GFP Polyclonal | ThermoFisher | A-11122 | 1/1000 |
| Antibody | β-Actin | Cell Signaling | 4967 | 1/1000 |
| Antibody | β-Tubulin | Millipore Sigma | Mab3408 | 1/1000 |
| Commercial assay or kit | Pyruvate Assay Kit | abcam | ab65342 | |
| Chemical compound | FM 1–43 Dye | ThermoFisher | T3163 | 4 µM |
| Other | Primer: Human Phosphofructokinase P (PFKP) | ThermoFisher | Assay ID Hs00242993_m1 | |
| Other | Primer: Human Phosphofructokinase M (*PFKM*) | ThermoFisher | Assay ID Hs00175997_m1 | |
| Other | Primer: Human *G6PD* | ThermoFisher | Assay ID Hs00166169_m1 | |
| Other | Primer: Human GAPDH | ThermoFisher | Assay ID Hs99999905_m1 | |

### *Drosophila* genetics

Standard crosses were incubated at 25°C with a 12 hr alternating dark-light cycles. (i) w[1118] (ii) w[1118]; UAS-TDP43[WT]-YFP (iii) w[1118]; UAS-TDP43[G29S8]-YFP (iv) UAS-GLUT3 flies were crossed with the GAL4 motor neuron driver D42. GLUT3 TDP-43 recombinant flies were generated using standard genetic techniques. GLUT3 flies were described in *Besson et al. (2015)*.

TDP-43 was expressed in motor neurons and glia using the D42 GAL4 (*Gustafson and Boulianne, 1996*) and repo GAL4 (*Sepp et al., 2001*) drivers, respectively. w[1118] was crossed with either the D42 or repo GAL4 drivers as a control. Endogenous TDP-43 was knocked down via RNAi using y[1] v [1]; P{y[+t7.7] v[+t1.8]=TRiP.HMS01932}attP40 and y[1] v[1]; P{y[+t7.7] v[+t1.8]=TRiP.HM05194} attP2.

To manipulate *PFK*, we used w[1118]; P{w[+mC]=UAS Pfk.T}3 for over expression, and y[1] sc* v[1]; P{y [+t7.7] v[+t1.8]=TRiP.HMS01324}attP2 or y[1] sc* v[1]; P{y[+t7.7] v[+t1.8]=TRiP .GL00298}attP2 (Line 2, viable only in glial expressing flies) for PFK knock down. *attP2* was used as a genetic background control for PFK RNAi knock-down experiments. All lines were obtained from the Bloomington Stock Center.

*PFK* RNAi efficiency was assayed by crossing *PFK* RNAi to D42 GAL4. Ventral nerve cords were dissected and analyzed via qPCR (see below for primers). A Welch's t-test was used to identify significance.

## Fly food supplementation

Standard yeast/cornmeal/molasses food was heated and then allowed to cool to 55–60°C. For high sugar supplementation, additional glucose was added to reach a final concentration of 357 g/L (10X compared to regular food). Supplemented food was dispensed into vials and allowed to cool.

## Locomotor assays

Larval turning assays were performed as previously described (*Estes et al., 2011*; *Estes et al., 2013*). Briefly, larvae are placed on a petri dish filled with solidified agar (2.5%) and grape juice. Third instar larvae are transferred from a vial onto the agar/juice surface and are allowed to acclimate. Using a clean paintbrush, larvae are turned ventral side up. The time it takes for larvae to turn themselves ventral side down and make a forward motion is recorded. Larval turning data were analyzed using Kruskal-Wallis multiple comparison tests (GraphPad Prism v8). Negative geotaxis assays on adult flies were performed by using TDP-43 expressing flies with a milder phenotype/lethality. TDP-43[WT] and TDP-43[G298S] flies were previously described in *Khalil et al. (2017)*. Briefly, TDP-43 and GLUT-3 were expressed in neurons with the pan neuronal driver, elav GAL4. Female flies of 11 day old were anesthetized with $CO_2$ and placed in a plastic column (diameter 1.3 cm) with a mark at 22 cm. After 20 min recovery, flies were gently tapped to the bottom of the column. Then the percentages of flies reaching the 22 cm mark or remaining at the bottom were counted after 60 s. The test was repeated three times and results were averaged. The data are the mean of at least six trials and are expressed as mean ± SEM. Statistical comparisons were performed using one-way ANOVA followed by Tukey's multiple-comparison test.

## Metabolite analysis

Metabolomic analysis was conducted by Metabolon, Inc as previously described (*Joardar et al., 2015*; *Manzo et al., 2018*). Briefly, 50–60 wandering third instar larvae (~50–60 mg) per sample were collected and flash-frozen in liquid nitrogen. A total of 5 replicates per genotype were collected. Samples were analyzed by ultrahigh performance liquid chromatography/mass spectrometry (UHPLC/MS), or by gas chromatography/mass spectrometry (GC/MS). One-way ANOVA was used to identify biochemicals that differed significantly between experimental groups.

## Pyruvate measurements

Pyruvate was measured using the Pyruvate Assay Kit (Abcam-ab65342) as per manufacturer's instructions. In brief, 20 3[rd] instar larvae per genotype were analyzed in three biological replicates. Standard curve was generated kit provided reagents. Results are shown (*Figure 1—figure supplement 1*) in nmoles/microliter.

RNA isolation, cDNA preparation and qPCR cDNA protocol has been previously described (*Manzo et al., 2018*). Briefly, five ventral nerve cords (VNCs) were isolated from third instar larvae per genotype using HL-3 saline buffer (70 mM NaCL, 5 mM KCl, 22 mM MgCl$_2$, 10 mM NaHCO$_3$, 5 mM trehalose, 115 mM sucrose, 5 mM HEPES, pH 7.3). RNA was extracted using the RNeasy RNA Extraction Kit (Qiagen) and stored at −80℃. Genomic DNA (gDNA) was cleared using a DNase I (Thermo Fisher Scientific) digestion protocol. Following gDNA clearing, reverse transcription was performed using the Maxima First Strand cDNA Synthesis Kit for RT-qPCR (Thermo Fisher Scientific). The quality of the cDNA was tested using GPDH primers (see below for sequence) and GoTaq end-point polymerase chain reaction (PCR) protocol (Promega). Quantitative PCR was performed using a SYBR Green protocol (Applied Biosystems) and primers designed to span exon-exon junctions. Each reaction was conducted in 3–6 biological replicates with three technical replicates each. ΔΔCT values are normalized to GPDH (*Pfaffl, 2001*).The Kruskal-Wallis test was used to assess statistical significance in GraphPad Prism v8. Protocols for human tissue RNA isolation and cDNA preparation have been previously described (*Bakkar et al., 2018*). RT-PCR was performed using TaqMan Gene Expression Assays following the TaqMan Fast Advanced Master Mix protocol (Applied Biosystems) on the QuantStudio 6 Flex instrument (Applied Biosystems).

## Primer sequences and taqman probes

*Drosophila Phosphofructokinase* (PFK): Forward primer - TGGACGAGCTGGTCAAGAAC; Reverse primer - CCACAAGAGCTAAATAGCCG
*Drosophila G6PD (Zw)*: Forward primer - AAGCGCCGCAACTCTTTG; Reverse primer - AGGGCGGTGTGATCTTCC
*Drosophila GPDH:* Forward primer - CCGCAGTGCTTGTTTTGCT; Reverse primer – TATGGCCGAACCCCAGTTG
Human *Phosphofructokinase P* (*PFKP*): Assay ID Hs00242993_m1
Human *Phosphofructokinase M* (*PFKM*): Assay ID Hs00175997_m1
Human *G6PD*: Assay ID Hs00166169_m1
Human GAPDH: Assay ID Hs99999905_m1

## Glucose sensor

Third instar larval brains where dissected in HL3-buffer (70 mM NaCl, 5 mM KCl, 20 mM MgCl2, 10 mM NaHCO3, 115 mM sucrose, 5 mM trehalose, 5 mM HEPES; pH 7.1). Of note, the glucose sensor is not sensitive to trehalose (*Volkenhoff et al., 2018*). Larvae were pinned down on a Sylgard plate using insect pins and cleaned so that the ventral nerve cord is clearly visible. A 40x water immersion lens was used to image larvae individually. For glucose stimulation, HL3-buffer was removed and replaced with 5 mM glucose supplemented HL3 (70 mM NaCl, 5 mM KCl, 20 mM MgCl2, 10 mM NaHCO3, 115 mM sucrose, 5 mM glucose, 5 mM HEPES, pH 7.1).

Fluorescent images were acquired using a Zeiss LSM880 NLO upright multiphoton/confocal microscope. CFP was excited using a 405 nm laser, and images were acquired in both the CFP (CFP 465–499 nm) and the FRET (535–695 nm) detection channels.

Data was analyzed using ImageJ with a drift correction plugin turboreg (*Thévenaz et al., 1998*). Regions of interest (ROIs) were manually selected and the mean grey value of each ROI was extracted. The mean grey values for the FRET and CFP channels was taken to calculate the FRET ratio (F/C), which corresponds to the intracellular glucose concentration.

Each ROI was imaged every 10 s for 10 min (baseline) and then for another 10 min under high glucose HL-3 (stim; see *Figure 3—figure supplement 1* for representative examples). Each FRET ratio was normalized to the mean baseline value for each genotype. For baseline vs stim comparisons (*Figure 3B*), the normalized FRET ratio of each ROI was taken from minutes 5–10 and minutes 15–20.

## Western blots

TDP-43 was detected via its YFP tag using either the Clontech/Takara Living Colors GFP monoclonal antibody (6326375) at 1:1000 (for *Figure 4—figure supplement 4*) or the ThermoFisher polyclonal antibody (A-11122) at 1:1000 (for *Figure 6—figure supplement 1*). For normalization we used either the Cell Signaling polyclonal rabbit anti β-Actin antibody (4967) at a 1:1000 dilution or the Millipore

Sigma monoclonal antibody against beta tubulin (Mab3408) at 1:1000. 5% milk was used as blocking agent for all westerns.

## FM1-43 experiments

Larvae are dissected in HL-3 buffer on a sylgard dish as previously described. FM1-43 dye uptake assays were conducted using 4 µM FM1-43FX, 90 mM KCl, and 2 mM Ca2+ for stimulation. Methods adapted from *Kuromi and Kidokoro (2005)* and *Verstreken et al. (2008)*. To determine dye uptake signal, the background-corrected fluorescence for every synaptic bouton was calculated by subtracting the mean background fluorescence of cell-free regions equal in size to the synaptic bouton.

## Bouton counts

Larval NMJs were dissected and fixed with 3.5% paraformaldehyde in PBS (pH 7.2: 130 mM NaCl, 5 mM $Na_2HPO_4$, 5 mM $NaH_2PO_4$) for 30 min. Antibody dilutions are as follows: 1:300 mouse anti-DCSP2 (DSHB), 1:500 anti-mouse Alexa Fluor 568 (Invitrogen), 1:200 Anti-HRP-CY5 conjugate (Invitrogen), and 1:40 Phalloidin-488 (Invitrogen). Blocking solution contained 2% BSA and 5% Normal goat serum. Methods have been previously described in *Coyne et al. (2014)*.

Larval muscles 6 and 7, segment A3, were imaged using an LSM 880 confocal microscope (Zeiss). NIH ImageJ was used to count bouton numbers and measure muscle area. Bouton count was normalized to muscle area to account for size variation.

## iPSC methods

Control iPSC and TDP-43 iPSC cells were differentiated to motor neurons as previously described (*Donnelly et al., 2013*; *Zhang et al., 2016*).

## Cellular fractionations

To generate cellular fractions, 25 Third instar wandering larvae were flash frozen, followed by homogenization in Trizolreagent using the Bullet Blender Bead Lysis Kit Green in conjunction with a Bullet Blender Blue 24. Homogenates were then centrifuged at 2000 x g for 3 min. The supernatant was separated from both the fat layer and the beads then it was further centrifuged at 25,000 g for 30 min. The resulting supernatant is the soluble fraction and the insoluble fraction is generated by resuspending the pellet in urea buffer (30 mM Tris, 7 M Urea, 2 M Thiourea, 4% CHAPS, 1X Protease Inhibitor Cocktail (Roche), 0.5 mM PMSF, 40 units/µl RNAsin plus, pH 8.5). RNA was isolated from inputs and the cellular fractions utilizing the manufactured described protocol for Trizol reagent. cDNA was then generated utilizing Thermoscientific Maxima First Strand cDNA Synthesis Kit for RT-qPCR. RT-qPCR was conducted on the StepOnePlus RT-qPCR machine utilizing Sybr reagents.

## Statistics

All statistical analyses for locomotor assays or qPCR comprising three or more data sets were conducted using Kruskal-Wallis with uncorrected Dunn's test in Prism v8.0 (GraphPad). When only two data sets were compared, either Welch's test or Wilcoxon signed-rank test were used. Biological replicates for qPCR and metabolomics are defined as independent sample collections per genotype, from independent genetic crosses. All qPCR experiments included three technical replicates per biological replicate. Outliers were removed using ROUT method (Q = 1%) in Prism v8.0.

## Additional information

### Funding

| Funder | Grant reference number | Author |
| --- | --- | --- |
| National Institutes of Health | T32GM008659 | Ernesto Manzo |
| Howard Hughes Medical Institute | Gilliam Fellowship for Advanced Studies | Ernesto Manzo |
| University of Arizona | Undergraduate Biology Research Program, Undergraduate scholarship | Abigail G O'Conner Dakotah D Shreiner |

| Arnold and Mabel Beckman Foundation | Undergraduate scholarship | Jordan M Barrows |
|---|---|---|
| Association pour la Recherche sur la Sclérose Latérale Amyotrophique et autres Maladies du Motoneurone | | Jean-Charles Liévens |
| Target ALS | | Robert Bowser |
| Target ALS | Post-Mortem Core | Robert Bowser |
| Barrow Neurological Foundation | | Rita Sattler |
| National Institutes of Health | NS091299 | Daniela C Zarnescu |
| Muscular Dystrophy Association | 418515 | Daniela C Zarnescu |

The funders had no role in study design, data collection and interpretation, or the decision to submit the work for publication.

## Author contributions

Ernesto Manzo, Conceptualization, Data curation, Formal analysis, Supervision, Funding acquisition, Validation, Investigation, Visualization, Methodology, Writing—original draft, Writing—review and editing; Ileana Lorenzini, Conceptualization, Formal analysis, Investigation, Methodology, Writing—original draft; Dianne Barrameda, Abigail G O'Conner, Benjamin E Rabichow, Investigation; Jordan M Barrows, Archi Joardar, Conceptualization, Investigation; Alexander Starr, Dakotah D Shreiner, Formal analysis, Investigation; Tina Kovalik, Investigation, Writing—original draft; Erik M Lehmkuhl, Formal analysis, Investigation, Writing—original draft; Jean-Charles Liévens, Conceptualization, Formal analysis, Funding acquisition, Investigation, Writing—original draft; Robert Bowser, Supervision, Funding acquisition, Project administration; Rita Sattler, Conceptualization, Formal analysis, Supervision, Funding acquisition, Project administration; Daniela C Zarnescu, Conceptualization, Resources, Data curation, Formal analysis, Supervision, Funding acquisition, Methodology, Writing—original draft, Project administration, Writing—review and editing

## Author ORCIDs

Ernesto Manzo (ID) https://orcid.org/0000-0002-1526-8144
Jordan M Barrows (ID) http://orcid.org/0000-0002-9693-0207
Robert Bowser (ID) http://orcid.org/0000-0001-5404-7259
Daniela C Zarnescu (ID) https://orcid.org/0000-0002-9607-0139

## Decision letter and Author response

Decision letter https://doi.org/10.7554/eLife.45114.036
Author response https://doi.org/10.7554/eLife.45114.037

# Additional files

## Supplementary files

• Supplementary file 1. Summary of carbohydrate metabolites in TDP-43$^{WT}$ and TDP-43$^{G298S}$ compared to $w^{1118}$ controls. Altered metabolites in third instar larvae crossed with the motor neuron driver D42 GAL4 were measured using gas or liquid chromatography followed by mass spectrometry. Red and green colored cells indicate statistically significant changes (P$_{value}$ <0.05) that are increased and decreased, respectively. Light red and light green colored cells indicate upward or downward trends, respectively (P$_{value}$ <0.1).
DOI: https://doi.org/10.7554/eLife.45114.031

• Supplementary file 2. Summary of demographic information for patient samples used to quantify *PFKP*, *PFKM* and *G6PD*.
DOI: https://doi.org/10.7554/eLife.45114.032

• Supplementary file 3. Summary of iPSC MNs used to quantify *PFKP*, *PFKM* and *G6PD*. Patient cell lines used for qPCR analysis are shown.
DOI: https://doi.org/10.7554/eLife.45114.033

• Transparent reporting form
DOI: https://doi.org/10.7554/eLife.45114.034

### Data availability

All data generated or analysed during this study are included in the manuscript and supporting files.

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
