## [Decision Letter]

Thank you for submitting your article "Glycolysis upregulation is neuroprotective as a compensatory mechanism in ALS" for consideration by eLife. Your article has been reviewed by three peer reviewers, one of whom is a member of our Board of Reviewing Editors, and the evaluation has been overseen by K VijayRaghavan as the Senior Editor. The reviewers have opted to remain anonymous.

The reviewers have discussed the reviews with one another and the Reviewing Editor has drafted this decision to help you prepare a revised submission.

Summary:

Alteration of cellular energetic metabolism in neurodegenerative disease is an area of considerable interest. In this very strong study, largely using *Drosophila* models of TDP-43 mediated pathogenesis, data are presented in support of the concept that enhancing glycolysis is neuroprotective. In addition to using a fly model, the investigators do provide data that supporting alterations in glycolysis in ALS patient tissue and iPSC-derived motor neurons.

Essential revisions:

1) Show quantification of gene and protein expression in the experiments where these controls are missing. Reviewers noted this is important to clarify that phenotypes reflect actions of mutant TDP-43 and not simply TDP-43 overexpression.

2) Add additional iPScs (to the extent possible – try to find one or two more cells lines) or increase number of experiments (replicates) in patients-derived cells- at least to account for batch to batch variation.

3) Provide evidence that the fly models used have TDP-43 pathology (proteinopathies) using IF and western blots.

---

## [Author Response]

Essential revisions:1) Show quantification of gene and protein expression in the experiments where these controls are missing.

We understand this comment as a question of whether mitigating conditions such as GLUT3 or PFK overexpression (OE) exert their action by altering TDP-43 levels. We now include western blot data for both GLUT3 and PFK OE showing that TDP-43 levels are unchanged in the context of either GLUT3 or PFK OE in motor neurons (see Figure 4 —figure supplement 4 and Figure 6—figure supplement 1). Therefore the mitigating effects on TDP-43 toxicity observed with GLUT3 and PFK OE likely occur through increased glucose availability and upregulated glycolysis, respectively, and not by altering TDP-43 expression.

Reviewers noted this is important to clarify that phenotypes reflect actions of mutant TDP-43 and not simply TDP-43 overexpression.We thank the reviewers for bringing up this important issue about TDP-43 models of proteinopathy. While overexpression of either WT or disease associated G298S causes similar phenotypes at the “whole organism level”, it is important to note that mechanistically, the two variants cause disease using distinct pathways. We have emphasized this issue in the text in several places throughout the Results and Discussion sections.2) Add additional iPScs (to the extent possible – try to find one or two more cells lines) or increase number of experiments (replicates) in patients-derived cells- at least to account for batch to batch variation.

We very much agree with this important point and were able to add a third differentiation to our iPSC experiments. All transcripts tested are now significantly altered (see Figure 2D and subsection “PFK transcript is upregulated in human tissue”). We were unable to obtain additional mutant TDP-43 mutant lines from ALS patients. This reflects the low frequency of TDP-43 mutations found in the patient population.

3) Provide evidence that the fly models used have TDP-43 pathology (proteinopathies) using IF and western blots.

We have previously showed using cellular fractionations and immunofluorescence that TDP-43 overexpression causes insolubilization of TDP-43 (Estes et al., 2013) and specific mRNAs (Coyne et al., 2015, Coyne et al., 2017).